# Explaining Age at Autism Spectrum Diagnosis in Children with Migrant and Non-Migrant Background in Austria

**DOI:** 10.3390/brainsci10070448

**Published:** 2020-07-14

**Authors:** Patricia Garcia Primo, Christoph Weber, Manuel Posada de la Paz, Johannes Fellinger, Anna Dirmhirn, Daniel Holzinger

**Affiliations:** 1Research Institute of Developmental Medicine, Johannes Kepler University, 4020 Linz, Austria; patricia.garcia_primo@jku.at (P.G.P.); christoph.weber@ph-ooe.at (C.W.); Johannes.Fellinger@bblinz.at (J.F.); Daniel.Holzinger@bblinz.at (D.H.); 2Department for Inclusive Education, University of Education Upper Austria, 4020 Linz, Austria; 3Institute of Rare Diseases Research (IIER) & CIBERER, Instituto de Salud Carlos III, 28029 Madrid, Spain; 4Institut für Sinnes- und Sprachneurologie, Konventhospital Barmherzige Brüder, 4020 Linz, Austria; Anna.Dirmhirn@bblinz.at; 5Division of Social Psychiatry, Medical University of Vienna, 1010 Vienna, Austria; 6Institute of Linguistics, Karl-Franzens University of Graz, 8010 Graz, Austria

**Keywords:** migration, autism, diagnosis, Europe, health system

## Abstract

This study explored (i) differences in age at Autism Spectrum Disorder (ASD) diagnosis between children with and without a migrant background in the main diagnostic centre for ASD in Upper Austria (ii) factors related to the age at diagnosis and (iii) whether specific factors differed between the two groups. A retrospective chart analysis included all children who received their first diagnosis before the age of 10 years (*n* = 211) between 2013 and 2018. Children with a migrant background were diagnosed 13 months earlier than those without (r = 0.278, *p* < 0.001), and had more severe delays in language, more severe autism, no Asperger’s syndrome, lower parental educational level and more frequent referrals by paediatricians. For the total sample, expressive language delay, severity of restricted and repetitive behaviours, higher nonverbal development, and paediatric referrals explained earlier diagnoses. There was a stronger effect of parental education and weaker effect of language impairment on age at ASD diagnosis in children with a migrant background. In conclusion, no delay in diagnosing ASD in children with a migrant background in a country with universal health care and an established system of paediatric developmental surveillance was found. Awareness of ASD, including Asperger’s syndrome, should be raised among families and healthcare professionals.

## 1. Introduction

(1) Prevalence of Autism Spectrum Disorder (ASD): The prevalence of ASD has risen in the last decade, with current rates of 1/59 in North America [1] and similar rates reported in Europe [2,3,4,5]. In the absence of ASD prevalence studies in Austria, epidemiological studies must be consulted [6], which would indicate a number of approximately 80,000 individuals with ASD diagnosis in this country—clearly a number that has a considerable impact on the national health and social systems.

*ASD and migration.* A growing number of European studies, particularly from Nordic countries, suggests an increased frequency of autism in children of immigrant parents [7,8,9,10]. However, a recent systematic review of ASD prevalence and migration status in Europe has shown no simple, linear relationship between prevalence or risk of ASD and immigrant status [11,12]. Earlier data from North American studies tend not to support this finding [13,14], whereas a recent ADDM (Autism and Developmental Disabilities Monitoring Network) study has shown a higher ASD prevalence in black children as compared to their white peers [15]. In Europe, the number of immigrants is substantial and also on the rise. In 2018, 22.3 million people with citizenship of a non-member country (4.4% of the EU-28 population) resided in an EU member state. Additionally, 17.6 million people in the EU were citizens of another EU member state [16]. Within the EU, Austria has one of the highest proportions of migrants (23.3%) [17]. The proportion of children who do not use German as their primary family language is 26% across all school types and 31% in primary schools [18].

(2) Age at ASD diagnosis and predictors: The increase in ASD prevalence has been accompanied by a decrease in the age at diagnosis in many countries [11,19], although there are exceptions [9]. Early ASD diagnosis is the gateway to support services [20] and interventions. Growing evidence points to the beneficial effects of enrolment at younger age in autism intervention on (a) the child’s long-term outcomes [4,21,22], (b) the family’s coping skills [23,24] and (c) costs to society [25,26]. Timely access to ASD diagnosis and support requires equitable access to health services, which is also one of the principles of the World Health Organisation (WHO) for universal health coverage [27]. Specific factors that affect the age at ASD diagnosis have been identified in several studies and are mainly related to clinical ASD presentation [28]. Severe language delay and more severe repetitive behaviours have been shown to be very common symptoms that prompt early medical consultation and diagnosis at a younger age. Hearing impairment and intellectual disability [28,29] have been found to delay identification of ASD in many cases. However, a low level of intellectual functioning can also stimulate earlier developmental investigation and earlier autism diagnosis [30,31,32]. Particular characteristics that are related to the child’s family such as living in under-resourced or rural areas [28], low parental occupational level [33] and/or lower socioeconomic status [34,35] have more recently been identified as factors related to later ASD diagnosis. Furthermore, characteristics of the health system (mainly presence or absence of universal coverage), and whether it has a tradition of surveillance or screening programmes, must be considered.

(3) Migration and age at diagnosis: Lastly, migration status may also be a cause of delayed diagnosis [28,36,37,38,39,40,41]. Several studies in the USA have shown that children from immigrant families are more likely to be diagnosed later (mostly after the age of 4) than their non-migrant peers [28,33,37,39,41]. Although these data show an association between parental immigration and delay of autism diagnosis in North America, a limited number of studies has investigated this outside the United States [42]. To our knowledge, in Europe only one study in the Netherlands has explored this research question; the authors concluded that the immigrant sample did not differ in age at ASD diagnosis from the non-immigrant sample [43]. The following specific factors related to the age at ASD diagnosis in children from immigrant families have been reported [44]: firstly, a lack of knowledge of the host country language can restrict access to ASD awareness campaigns and education about ASD [44,45,46,47]. In addition, culture-related differences, such as parental feelings of shame and guilt related to child disability, have been described between families of children with developmental disabilities and the values and expectations of service providers [48,49,50]. A lack of understanding of the host country’s health care system and community stigma can also prevent parents from seeking care [49,51]. Negative perceptions of services and a poor understanding of rehabilitation concepts can also delay access to specialised health services, and parents with a migration background have reported perceived discrimination by service providers [52]. Furthermore, culturally shaped interpretations of child behaviours (e.g., social communication) can influence the parents’ seeking of medical advice [53]. As a consequence, professional organisations such as the American Psychological Association increasingly emphasise the importance of accounting for cultural differences in both professional practices and research [54].

(4) Health Care Services for Children with ASD in Upper Austria: The Austrian federal state of Upper Austria (1.48 million inhabitants) has a high proportion of immigration. In Upper Austria, 31% of all primary school children have a family language other than German [18]. The Austrian federal state of Upper Austria provides an extensive network of healthcare institutions and a universal-coverage health insurance system. The social system in Austria offers free public early intervention for all children with developmental delays. In Upper Austria, a paediatric well-baby check-up program is provided free of charge from birth to school entry (at the age of 6 years) and is well accepted by families. An important element of this programme is a specific language screening programme (SPES) that was developed in 2007 by the Institute of Neurology of Language and Senses (ISSN) to identify children with increased risk for persisting language difficulties at the 2- and 3-year paediatric check-ups. The SPES language screening is used on a voluntary basis by the majority of paediatricians. Children who fail the short assessment of receptive vocabulary are referred for further neurodevelopmental assessment. By use of this screening the age of autism diagnosis could be significantly decreased within the last decade. About 50% of all children live in rural areas (i.e., thinly populated areas; 12% of these children do not use German as their primary family language), 13% live in the city of Linz, where the proportion of children with a family language other than German is 56% [55].

The out-patient clinic of the ISSN, situated within a public general hospital in the city of Linz, has been the major clinical focus for developmental disorders for the local community in Upper Austria in the last decade. It is the main centre where the complex multidisciplinary diagnostic evaluation needed for ASD diagnosis is regularly conducted [56]. In the associated Autism Centre, ASD-specific early intervention—following primarily the Early Start Denver Model—is provided [57]. In the last decade, the percentage of patients with a migrant background using the diagnostic clinic has risen from about 40% to 50%. Thus, data from this clinic’s medical charts lend themselves to investigating factors that may determine a possible relationship between immigrant status and access to diagnosis.

### Aim and Hypotheses

This study aimed to explore (i) whether there are differences in terms of age at ASD diagnosis between children with or without a migrant background at the principal diagnostic centre for ASD in Upper Austria, (ii) what factors might be correlated with the age at diagnosis of these children and (iii) whether predictors of age at diagnosis differ between children with and without a migrant background.

## 2. Methods

### 2.1. Study Design

A cross sectional study was used. All children of parents with and without a migrant background (see definition below) who attended the diagnostic centre of the ISSN in the study period (01-01-2013 to 31-12-2018) and received an ICD-10- diagnosis (International Classification of Diseases, 10th Revision [58]) of ASD (F84) for the first time before the age of 10 years were considered in the study.

Migration background cannot be sufficiently defined by citizenship, since there are many families with parents born outside Austria (and even more second-generation families) who have Austrian citizenship. The ISSN medical records did not contain systematic information on the parents’ and children’s places of birth, but detailed information about the language(s) spoken in the family. Therefore, a family was considered to have a migrant background if both parents (or one parent in a single-parent family) used a language other than German as their primary language in the family. There were no members of Austria’s autochthonous ethnic minorities in the study sample.

Age of diagnosis was limited to a maximum of 10 years for both samples, since there were only few individual children in our clinical population who received an ASD diagnosis after that age. All ASD cases were evaluated at the ISSN by a developmental paediatrician, a clinical psychologist and a clinical linguist using various standardised tests depending on each child’s developmental level and a clinical (non-standardised) interview. The ICD-10 [58] was used as a basis for clinical diagnosis of ASD and for the classification codes in the reports. All clinicians had been trained in the implementation and scoring of the ADOS-2 [59], which was applied in the majority of cases. Main outcomes were summarised in a comprehensive medical report.

### 2.2. Study Variables

Dependent variable: chronological age of the child at first ASD diagnosis. Independent variables: grouped by (a) sociodemographic, (b) clinical characteristic and (c) referral to diagnosis.

(a) Sociodemographic characteristics: Migration background status was based on language(s) predominantly spoken by the parents in the family as reported by the parents themselves. A family was considered to have a migrant background if both mother and father (or a single parent) used a language other than German as their primary language at home. The languages reported as being used at home by the study population were then categorised into five geographical regions: South-Eastern and Eastern Europe, Western and Northern Europe, Middle and East Asia, West Asia, and others (see Appendix A). No parent was excluded due to insufficient language skills since translators were always available within the health system. Parental educational level was derived from the information provided by the parents in the registration form prior to clinical evaluation. Parents reported their current occupational level and/or educational background, and this information was then categorised into two levels of education: with or without high school diploma. For the analysis the highest parental educational level (i.e., of either father or mother) was used. Location of family residence was also elicited from the information provided by the parents in the registration form. This data was used to determine the degree of urbanisation of the place of residence (densely populated urban areas vs. thinly populated areas) and to estimate the distance between family residence and diagnostic clinic (in kilometres).

(b) Clinical characteristics: Gender was extracted from the medical record of the child. The non-verbal IQ score is the standardised non-verbal quotient of the child at the time of ASD diagnosis, either extracted directly from the evaluation report or calculated using the well-known quotient formula with the non-verbal developmental age of the child divided by the chronological age indicated in the report. The most common standardised cognitive tests reported were Mullen Scales of Early Learning (MSEL) [60], the Bayley Scales of Infant Development III [61] and the Hamburg-Wechsler Intelligenz test IV [62], depending on the child’s age and level of functioning. Language Scores (Receptive and Expressive): receptive (RL) and expressive (EL) language developmental quotient scores, usually assessed by use of the MSEL, were either obtained directly from the diagnostic report or calculated from age-equivalent scores (age-equivalent scores/chronological age × 100) when the clinician indicated the language level of the child in months (usually based on a standardised language test, but also according to clinical picture when administration of a test was not possible due to the characteristics of the child) (PGP and DH). Language skills where assessed by linguists experienced in multilingualism in the child’s primary language. If necessary, interpreters were used and parental observations were taken into consideration. Children with a developmental quotient below 50 were considered to have a moderate or severe delay. Variables of ASD Subtype: ICD-10 Code diagnosis of ASD (F84) ADOS Calibrated Severity Scores (CSS) for Social Affect (SA) and Restricted and Repetitive Behaviours (RRB) were transformed as appropriate from the original SA and RRB raw scores of the clinical report, following the indications of Gotham et al. [63], based on the child age, module and/or version of ADOS administered and reported in the clinical record (PGP and DH). A change in the clinical diagnostic procedure within the study period (from ADOS-G to ADOS-2) [64] was taken into account by the calibrated severity scores.

(c) Referral to diagnosis was either directly stated by the parents in the registration form or reported to the clinician at the evaluation (and therefore included in the diagnostic report).

### 2.3. Procedures

The clinical charts of all cases were revised and abstracted by researchers with considerable knowledge and experience in the field of neurodevelopmental disorders. To ensure accuracy, reliability and consistency in data abstraction, we developed, tested and revised a Record Review Protocol (RCR) following best practice [65,66]. An initial list of variables to be captured was tested in a small pilot study, discussed by the research team, and adjusted where necessary. In the case of conflicting or uncertain information in the medical charts, consensus decisions (PGM, DH) were made. The information that was reviewed referred to demographic and clinical characteristics of the children and their families and to the system of referral for ASD diagnosis.

Ethics: This study was approved by the ethics committee (Ethikkommission der Medizinischen Fakultät der Johannes Kepler Universität), Nr. 1140/2020 Version 3, following the rules of the Declaration of Helsinki of 1975 revised in 2013 [67].

### 2.4. Statistical Analysis

First, we separately calculated descriptive statistics (means, standard deviations for continuous variables and percentages for categorical variables) of all study variables for children with and without a migration background. Second, bivariate correlations were calculated between the age at diagnosis and the sociodemographic and clinical variables. This was done for the total sample and also separately according to migration status. In order to test for differences in correlations between the migrant and non-migrant groups, a Wald χ^2^-test was carried out. Correlations were also calculated for different age groups in order to identify possible age-dependent predictors of the age at diagnosis. For all bivariate analyses we calculated effect size measures in a correlation metric, more specifically, Pearson’s r for the association between continuous variables, point biserial correlation for the association between binary variable and continuous variables and phi or Cramer’s V for the association between categorical variables. Finally, we used linear regression models to evaluate the effects of all predictors simultaneously. We also tested whether the association between predictors and age at diagnosis differed by migrant status. This was achieved by including interaction terms (e.g., migrant × sex) in the regression models [68]. We report unstandardised regression coefficients (b) and their standard errors (SE) and standardised coefficients (β) that have the same metric as correlation coefficients and could likewise be interpreted as effect size measures [69,70]. Several study variables had a high number of missing values (e.g., 42% missing values for parental education; see also Table 1). Preliminary analysis indicated that missingness depended on other variables (i.e., missing at random [71]). For example, children with missing values for parental education were diagnosed later (M = 52 months vs. M = 44 months). Thus, in order to avoid bias and a loss of statistical power, we used multiple imputation (MI)—a state-of-the-art technique for dealing with missing data—to replace missing values [71]. Firstly, the missing data was imputed by available data, and several imputed data sets were generated. Secondly, analyses were carried out for all data sets, and lastly the results of all data sets were pooled and final coefficients and standard errors were computed according to Rubin’s combination rules [72]. Specifically, we used the blimp software [71,73], which implements a multiple imputation by chained equations (MICE) algorithm. All study variables were used for the imputation models, and imputations were carried out separately for the migrant and non-migrant groups. In order to obtain accurate standard errors, 200 imputed data sets were generated [74]. M*plus* 8.2. [75] was used for all analyses based on imputed data. Due to the non-normal distribution of the dependent variable of age at diagnosis (skewness = 1.411, kurtosis = 1.424), a maximum-likelihood estimation with robust standard errors (MLR) was used. Only descriptive results are reported for non-imputed data. These analyses were carried out using SPSS 26 and Jamovi [76].

## 3. Results

### 3.1. Descriptive Results

In Table 1 the descriptive results (based on non-imputed data) both for the total sample and for the non-migrant and migrant subsamples separately are shown. Results based on imputed data are provided in the Appendix A.

Of the 211 children in the sample, 58% received a diagnosis of autistic disorder, 10% of Asperger’s disorder, and 32% of Pervasive Developmental Disorder–Not Otherwise Specified (PDD-NOS). Children with autistic disorder were diagnosed at an average age of 39.4 months, followed by children with PDD-NOS (50.8 months) and children with Asperger’s disorder (75.9 months). Notably, children with Asperger’s disorder were almost exclusively from the non-migrant group; only one case in the children with migrant background subsample had received an Asperger’s disorder diagnosis. The difference in the ICD-10 ASD type between the non-migrant and migrant groups is highly significant (Cramer’s V = 0.385, *p* < 0.001).

The mean age at diagnosis was 46.7 months (SD = 22.8) for the total sample. Values ranged from 12 to 119 months. The non-migrant group was diagnosed at a mean age of 54 months as compared to 41.2 months for the migrant group. Thus, children with a migration background received their autism diagnosis significantly earlier (difference of about 13 months) than those without (r = 0.278, *p* < 0.001). The distributions of the age at diagnosis for the total sample and the migration-status subsamples are shown in Figure 1. The sample with a migrant background is characterised by a reduced distribution with a stronger concentration around ages 2–4 and a smaller proportion of children diagnosed with ASD at a later age.

### 3.2. Bivariate Analysis

In the total sample, there are moderate correlations between age at diagnosis and language scores. Children with lower expressive (r = 0.423, *p* < 0.001) and receptive (r = 0.355, *p* < 0.001) language scores were diagnosed significantly earlier. There is also a moderate correlation between RB-CSS and age at diagnosis (r = −0.370, *p* < 0.001). Children with higher RB-CSS received their diagnosis significantly earlier than those with lower RB-CSS. Similarly, analysis revealed a weak correlation between SA-CSS and age at diagnosis (r = −0.158, *p* < 0.05); more specifically, diagnosis occurred earlier in the case of higher severity scores for social affect. Finally, children who were referred by a paediatrician received their diagnoses earlier (r = −0.255, *p* < 0.001, M 40.78 months) than the rest of the sample (M 51.16 months). Table 2 shows the correlations between age at diagnosis and predictor variables for the total sample and the migration-status subsamples.

There are some notable differences between the strengths of the correlations with age of diagnosis between children with and without a migrant background. First, for migrant-background children there is a negative association between age at diagnosis and parental education (r = −0.169, *p* < 0.05). Thus, children of parents with migrant status who had at least graduated high school were diagnosed at an earlier age than children of parents with migration background who had no high school diploma. In contrast, this association is of comparable strength but of opposite direction and not statistically significant in the non-migrant subsample (r = 0.198, *p* > 0.05). Second, the correlations between both ELQ and RLQ and age at diagnosis are of moderate strength and statistically significant for the non-migrant group (ELQ, r = 0.443, *p* < 0.001; RLQ, r = 0.341, *p* < 0.001), but weaker and non-significant for the migrant group (ELQ, r = 0.216, *p* < 0.10; RLQ, r = 0.177, *p* > 0.05). Finally, it is interesting to note that the correlation between SA-CSS and age at diagnosis is significant for the migrant subsample (r = −0.263, *p* < 0.01), but not for non-migrant subsample (r = 0.011, *p* > 0.05). However, these two correlations do not differ significantly.

Table 3 presents the results of exploring age effects on the correlations between the predictors and age at diagnosis for two subsamples defined by age at diagnosis (<48 months vs. ≥48 months) and migration status. Note that these analyses suffer somewhat from small sample sizes (e.g., *n* = 23 for the migrant group in the “age at diagnosis ≥ 48 months” subsample). Nonetheless, there are several interesting results. First, there is no significant correlation between age at diagnosis and ELQ and RLQ, respectively, in the younger subsample, neither for children with nor for those without a migrant background. These correlations are only significant for non-migrant-background children in the older group (ELQ, r = 0.462, *p* < 0.001; RLQ, r = 0.346, *p* < 0.05). Thus, non-migrant-background children aged ≥ 48 months at diagnosis received their diagnoses earlier, the lower their language scores were. A somewhat similar pattern—although individual correlations are not significant—was found for the correlations between RLQ and age at diagnosis in the migrant group. RLQ and age at diagnosis correlate moderately (r = 0.332; *p* < 0.10) for migrant-background children aged ≥ 48 months at diagnosis, but not for those who received their diagnosis before 48 months (r = −0.122; *p* > 0.05). Second, the RB-CSS is only correlated with age at diagnosis for children who received their diagnosis at the age of 48 months or later. In detail, there are moderate to strong correlations between RB-CSS and age at diagnosis for children both with (r = −0.530, *p* < 0.01) and without (r = −0.477, *p* < 0.001) a migrant background. In the subsample of children who were younger at diagnosis, these correlations are much smaller and not significant (correlation differences between age subsamples are significant). Finally, there is a negative and significant small correlation between age at diagnosis and distance between home and hospital in the older non-migrant subsample (r = −0.282, *p* < 0.05). Thus, children from this subsample received their diagnoses later, the closer they lived to the hospital. For children with a migrant background in this age group, the correlation is not significant and positive (r = 0.179, *p* > 0.05). The correlations differ marginally (*p* < 0.10).

### 3.3. Regression Analysis

The results of the regression models are shown in Table 4. Due to the high correlation between ELQ and RLQ (r = 0.84), a unique composite language score (Language Quotient, LQ = (RLQ + ELQ)/2) was used in the regression models. All 10 predictors explain 34.6% of the variance in age at diagnosis. In accordance with the bivariate results, the LQ turned out to be the best predictor (β = 0.389, *p* < 0.001), followed by RB-CSS (β = −0.301, *p* < 0.001) and referral by paediatrician (β = −0.206, *p* < 0.01). Interestingly, the association between IQ and age at diagnosis, which was not significant in the bivariate analysis, is negative and statistically significant (β = −0.175, *p* < 0.05), when we control for other predictors. In fact, the association becomes significant as soon as we control for LQ; thus, given a constant LQ, the higher the IQ, the younger the child at diagnosis.

The regression model shows that migrant status is only marginally significantly associated with age at diagnosis (β = −0.136, *p* < 0.10) once other predictors have been controlled for. In detail, the difference in age at diagnosis, which amounted to about 13 months in the bivariate case (see Table 1), decreases to 6 months (b = −6.228) when other predictors are being controlled for. Again, the LQ is primarily responsible for differences between the bivariate and the multivariate analyses. Once we control for LQ, the difference between migration status and age at diagnosis decreases to roughly 7 months (b = −6.686, *p* < 0.05). Thus, about half of the difference in the age at diagnosis between the migrant and non-migrant groups is due to lower language scores of the migrant subsample.

In order to analyse possible differences in the associations between age at diagnosis and the independent variables between the migrant and non-migrant groups, we applied a series of regression models including interaction terms (for detailed results see Appendix A). None of the interaction effects were significant. Thus, in contrast to the bivariate results (Table 2), no differences between the migrant and non-migrant subsamples in the associations between age at diagnosis and independent variables were found in the multivariate models.

Finally, some further analyses were performed to better understand the negative IQ effect in the multivariate regression model. In detail, we included an interaction term between IQ and LQ, which turned out to be statistically significant (b = 0.007, *p* < 0.01). In order to interpret this interaction, we plotted the simple slopes, that is, the associations of IQ and age at diagnosis for both high (=M + SD) and low (=M – SD) levels of LQ. Figure 2 shows that there is no association between IQ and age at diagnosis for children with high LQ. This association is negative and significant at low LQ levels. Thus, children with high IQ and low LQ were diagnosed the earliest.

## 4. Discussion

The aim of this study was to investigate and compare age at diagnosis of autism spectrum disorders in children with or without a migrant background from an intake population of an outpatient clinic for developmental disorders in Austria and to examine the impact of sociodemographic variables, child characteristics and referral factors on age at diagnosis. We carried out a retrospective chart analysis including all children who received an ASD diagnosis before the age of 10 years for the first time.

Notably, our results demonstrate a higher percentage of children with (57%) than without (43%) a migrant background being diagnosed with ASD in our clinic. This contrasts with a reverse ratio of about 30:70 of primary non-German to German family language in Austria. On the one hand, this might be explained by the high percentage of immigrant families (56%) in the city of Linz and a closer distance to the diagnostic centre facilitating service use. On the other hand, the high percentage of children with migrant background might also be due to a higher prevalence of ASD in migrant populations, as described by some of the European studies [8,10]. Additionally, clinical experience shows a high uptake of medical referrals among families with a migrant background. In our sample, a significantly higher number of migrant- than non-migrant-background families reported that their child had been referred to the clinic by a paediatrician. There also seems to be a general tendency for families with a migrant background to use primary paediatric care more often and longer [77] than non-migrant-background families; the latter are more likely to consult general practitioners, who are often less well trained in identifying developmental disorders. Finally, the large proportion of ASD diagnoses in children with a migrant background is probably also related to free access to health care in Austria. Studies in countries without universal health care have shown that children with ethnic and racial differences are less often diagnosed with ASD [78].

Our main finding is that for the total study sample, children with a migrant background received the ASD diagnosis significantly earlier than the rest of the sample. However, in the younger subsample, diagnosed before 48 months of age, the mean age at diagnosis was almost identical for the migrant and non-migrant groups (34.4 and 34.6 months, respectively). For the total sample, the significantly younger mean age at diagnosis in the subsample with a migrant background is due to a lack of children with less severe autism symptomatology (Asperger disorder), who are usually presented to the clinic at an age older than 4 years. In any case, contrary to findings in other health systems [78], there are no disparities in early access to diagnosis of ASD in Austria, where free health care and a medical surveillance system (preventive medical check-ups) are provided for all.

In accordance with the literature [8,79,80], we found significant differences in clinical symptomatology between migrant- and non-migrant-background children diagnosed with ASD. The sample with a migrant background demonstrated significantly longer delays in language as well as more severe autism and tended to be more delayed in their nonverbal cognitive development. The differences in language development persisted after reducing the sample to those diagnosed before the age of 4 years. A comparison of the distribution of autism subtypes (according to ICD_10) showed an almost complete absence of diagnoses of Asperger’s disorder in children with a migrant background (*n* = 1) in contrast to the non-migrant subsample (*n* = 21). This accords with the results of Lehti et al. in 2015 [10], who found in a national birth cohort study of children diagnosed with Asperger’s disorder in Finland a significantly decreased likelihood of diagnosis in children whose parents were immigrants (adjusted odds ratio 0.2, 95% CI 0.1–0.4). The higher likelihood of childhood autism versus Asperger’s disorder syndrome in children with a migrant background might be related to specific risk factors associated with childhood autism (but not with Asperger’s disorder) that occur more frequently in the migrant population, such as perinatal complications, low birth weight and lower gestational age [9,81]. However, we strongly suspect that our results also reflect service utilisation. Underdiagnoses might be due to less awareness of milder forms of ASD in the migrant-background subsample with a significantly lower level of parental education. In addition, the culturally shaped appearance of Asperger’s disorder and communication problems between health professionals and parents and children might be other reasons for a lower number of this type of diagnosis.

Bivariate and multivariate analyses that identified factors correlated with delayed age at diagnosis for the total sample demonstrated a strong impact of the severity of language problems (expressive and/or receptive) on earlier ASD diagnosis. Severe delays in expressive language are easily noticed both by parents and by professionals in health and education systems and have been described as the most frequently observed symptoms of ASD that prompt medical consultation [82]. They have also been shown to be an important marker of non-apparent (invisible) developmental disorders [83]. In concordance with the findings of other studies, the severity of restricted and repetitive behaviours is another factor significantly related to earlier diagnosis of ASD, which is most likely a consequence of their obvious character compared to the often more subtle peculiarities in social communication. In addition, RBBs are often perceived as a cause of stress by the families and might therefore lead to external help being sought. The third-strongest factor related to earlier diagnosis was referral by a primary care paediatrician. Despite possible challenges in providing language- and culturally appropriate screening [78], well-trained paediatricians play an important role in reducing barriers to (early) ASD diagnosis. Finally, higher levels of intellectual functioning were found to be correlated with younger age at diagnosis, especially in children with seriously delayed language development, but it is also seen with average language scores. A severe language delay combined with significantly higher cognitive development may be even more obvious to parents and referring primary care professionals than a generalised developmental delay associated with a stronger need for explanation. With an expected majority of male gender in the sample, gender was not found to correlate significantly with age at diagnosis. Interestingly, parental education and the distance between the family’s place of residence and the diagnostic centre did not add significantly to explaining the variance in age at ASD diagnosis.

A comparison of factors related to age at identification of ASD in the migrant and non-migrant samples mostly showed significantly stronger effects of parental education in the group with a migration background. However, the correlation was small. In migrant-background families, educational disparities are often linked to pronounced deficits in (spoken and written) language that might have a particularly strong effect on access to health information.

The severity of (primarily expressive) language delay had significantly stronger effects on delayed ASD diagnosis in the non-migrant sample. This effect is almost exclusively observed in the older subsample (>4 years), where—due to the much higher number of children with Asperger’s disorder—variance in language development was significantly higher in the non-migrant sample. In school children with a migrant background, parents and professionals in health and education might ascribe language difficulties to bilingual language acquisition rather than to a developmental disorder. However, given the small size of the sample of older children with a migrant background (*n* = 23), results must be interpreted with caution. The minimal differences in factors explaining age at autism diagnosis between children with and without a migrant background are most likely a consequence of equitable access to medical care and a model of early identification with strong involvement of primary paediatric care that is well accepted.

The almost complete lack of significant effects of variables associated with age at diagnosis in the younger subsample is surprising at first glance. However, the variance in age at diagnosis was very small in the group aged younger than 4 years and probably related to the fact that paediatric screenings (mainly for expressive and receptive language) are usually offered at the ages of 2 and 3 years. Earlier referrals are highly unlikely. Furthermore, kindergarten, where autistic symptoms might be noticed, typically starts at the age of three. Long waiting periods for diagnostic services due to insufficient resources can cause inequalities and might be another reason that complicates the finding of significant predictors. The retrospective medical chart analysis presented has some limitations. Firstly, due to sample recruitment from a single clinic, effects of specific clinic-related factors on the age at diagnosis cannot be excluded. Secondly, another limitation of this study is its reliance on self-reported data about the parental level of occupation, since other information about Socioeconomic Status (SES), such as income or highest parental education, was not included in the medical charts. Even for information on parental occupation, data extracted from medical reports were very incomplete. Missing data, including those for other variables, were reported, and multiple imputations was used for missing values to avoid bias. Analyses of correlates of age at diagnosis by splitting the sample into two age subsamples and by migration status were interpreted with caution due to the small sizes of some of the subsamples.

## 5. Conclusions

This is one of the rare studies investigating age at diagnosis in children with a migrant background in Europe. Our findings highlight that for them the diagnosis of ASD is not delayed in Upper Austria, where universal health care includes a system of preventive medical check-ups provided mainly by paediatricians during the first years of life. Specific effects of lower parental education on age at diagnosis in families with a migrant background point to the importance of utilising existing preventive systems for systematic ASD screening and to the need to improve parent education on child development and health care services. The almost complete absence of diagnoses of milder forms of ASD (Asperger’s disorder) in children with a migrant background demonstrates the need for improving awareness of the whole autism spectrum by training professionals in health care and education and extending education to parents. Restricted and repetitive behaviours, language delay, and the combination of severe language delay with a relatively higher nonverbal development are possible symptoms of autism that can be used to identify ASD by observation by parents or professionals and by their inclusion in systematic screening tools.

For children both with and without a migrant background, referrals by primary care paediatricians significantly decreased age at diagnosis. A universal implementation of a streamlined model from systematic screening to timely diagnosis is expected to further reduce age at ASD diagnosis and to facilitate earlier access to specialised intervention.

## Figures and Tables

**Figure 1 brainsci-10-00448-f001:**
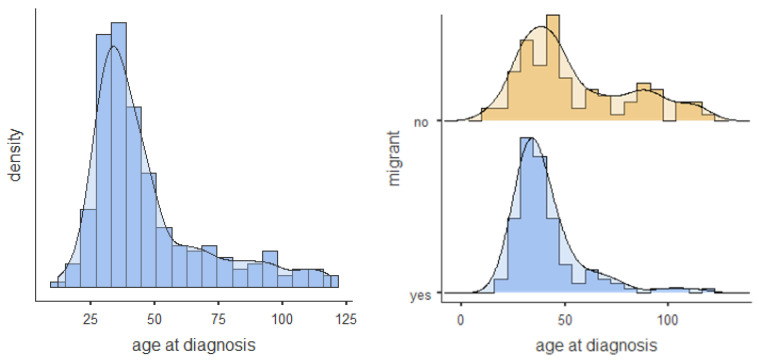
Age at diagnosis distribution histogram and density plot for total sample and split by migrant status.

**Figure 2 brainsci-10-00448-f002:**
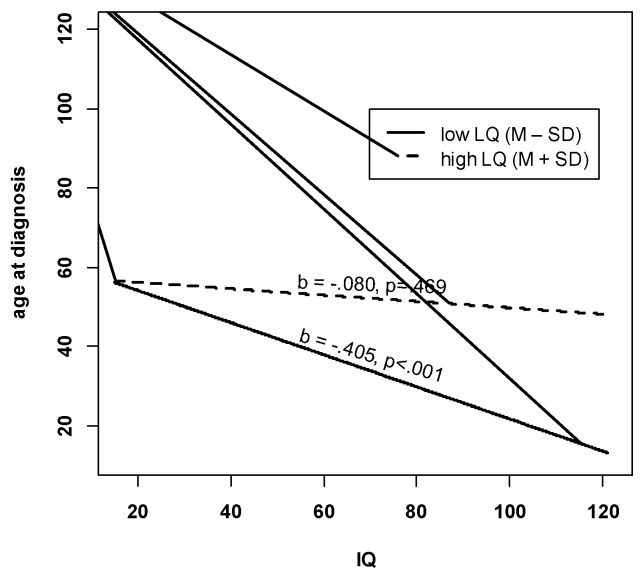
Interaction between LQ and IQ. Note. M = mean, SD = standard deviation. IQ = Non-Verbal Developmental Quotient. LQ = composite language score. b = unstandardised regression coefficient for IQ at high levels of LQ (M + SD) and low levels of LQ (M – SD).

**Table 1 brainsci-10-00448-t001:** Sample characteristics by migration status.

			Total (*n* = 211)	Non-Migrants (*n* = 91)	Migrants (*n* = 120)	Difference	
	Values	%M	M (SD) or n (%)	M (SD) or n (%)	M (SD) or n (%)	*ES r ^a^*	*p*
**Age at Diagnosis**	12–119 m.o.	0%	46.7 (22.80)	53.96 (26.86)	41.19 (17.33)	0.278	<0.001

**Sociodemographic Characteristics**							
Family Residence in Urban Area	No/Yes	0%	148 (70.1%)	51 (56.0%)	97 (80.8%)	0.268	<0.001
Distance to Hospital	0–187 km	0%	38.27 (37.50)	49.27 (41.67)	29.93 (31.73)	0.256	<0.001
Parental Level of Education above High School	No/Yes	41%	37 (30.1%)	23 (42.6%)	14 (20.3%)	0.241	0.007
**Clinical Characteristics**							
Male Gender	No/Yes	0%	174 (82.5%)	77 (84.6%)	97 (80.8%)	0.049	0.474
Non-Verbal Developmental Quotient, (D/IQ)	16–122	13%	62.26 (18.89)	65.5 (21.15)	59.67 (16.56)	0.155	0.036
D/IQ 50 or Below	No/Yes	13%	37 (20.1%)	16 (19.8%)	21 (20.4%)	0.008	0.915
Expressive Language Quotient (ELQ)	13–128	15%	46.38 (24.44)	57.42 (28.93)	39.34 (18.46)	0.340	<0.001
ELQ 50 or Below	No/Yes	15%	118 (65.9%)	37 (49.3%)	81 (77.90%)	0.297	<0.001
Receptive Language Quotient (RLQ)	15–133	15%	48.25 (24.65)	59.6 (29.04)	40.75 (18.23)	0.355	<0.001
RLQ Score 50 or Below	No/Yes	15%	116 (64.8%)	36 (47.4%)	80 (77.70%)	0.314	<0.001
ADOS-Calibrated Severity Scores—Total CSS							
Social Affect—CSS	1–10	18%	6.35 (2.19)	5.83 (2.16)	6.71 (2.15)	0.591.	0.041
Repetitive Behaviour—CSS	1–10	18%	5.85 (1.8)	5.76 (1.86)	5.91 (1.86)	0.009.	0.198
ICD-10 ASD Code							
Autism Disorder		0%	122 (57.80%)	38 (41.8%)	84 (70.00%)	0.385	<0.001
Asperger‘s Disorder			21 (10%)	20 (22%)	1 (0.80%)		
PDD-nos			68 (32.20%)	33 (36.2%)	35 (29.20%)		
**Referral to Diagnosis**							
Referred by Paediatrician	No/Yes	27%	76 (36%)	23 (25.3%)	53 (44.2%)	0.289	<0.001

Note. %M = percentage missing values, M = mean, SD = standard deviation. ^a^ Effect size estimates for differences between migrants and non-migrants in a correlation metric.

**Table 2 brainsci-10-00448-t002:** Correlations between sample characteristics and age at diagnosis separated by migration status.

	Total	Non-Migrants	Migrants	Correlation Difference between Groups ^a^
**Sociodemographic Characteristics**	r	r	r	*p*
Family Residence in Urban Area	−0.132	−0.073	−0.047	0.684
Distance Between Home And Hospital in Km	0.031	−0.050	−0.034	0.740
Parental Level of Education above High School	0.100	0.198	−0.169 *	0.035
**Clinical Characteristics**				
Male Gender	0.011	0.095	−0.048	0.252
Non-Verbal Developmental Quotient (IQ)	0.065	0.123	−0.131	0.136
Expressive Language Quotient (ELQ)	0.423 ***	0.443 ***	0.216	0.014
Receptive Language Quotient (RLQ)	0.355 ***	0.341 ***	0.177	0.059
ADOS-Calibrated Severity Scores—Total CSS				
Social Affect—CSS	−0.158 *	0.011	−0.263 **	0.206
Repetitive Behaviour—CSS	−0.370 ***	−0.446 ***	−0.326 ***	0.186
**Referred by Paediatrician**	−0.255 ***	−0.195	−0.220 *	0.707

Note. Correlations are based on multiple imputed data. ^a^
*p*-values for correlation differences between migrants and non-migrants are based on a Wald χ^2^-Test. *** *p* < 0.001, ** *p* < 0.01, * *p* < 0.05.

**Table 3 brainsci-10-00448-t003:** Correlations between sample characteristics and age at diagnosis according to migration status and age-at-diagnosis subsample.

	Age at Diagnosis < 48 Months		Age at Diagnosis ≥ 48 Months					
	(1) Non-Migrants (*n* = 50)	(2) Migrants (*n* = 97)	(3) Non-Migrants (*n* = 41)	(4) Migrants (*n* = 23)	Correlation Differences between Groups (*p*-Values) ^a^			
**Sociodemographic Characteristics**	r	r	r	r	1, 2	3, 4	1, 3	2, 4
Family Residence in Urban Area	−0.079	0.167	0.033	−0.114	0.244	0.597	0.700	0.461
Home Distance to Hospital in km	−0.009	−0.122	−0.282*	0.179	0.711	0.059	0.118	0.265
Parental Level of Education above High School	0.037	−0.094	0.167	−0.197	0.574	0.204	0.418	0.559
**Clinical Characteristics**								
Male Gender	0.090	0.034	−0.124	−0.200	0.680	0.848	0.355	0.204
Non-Verbal Developmental Quotient (IQ)	0.066	−0.150	0.168	0.235	0.305	0.989	0.371	0.213
Expressive Language Quotient (ELQ)	0.097	−0.172	0.462 ***	0.186	0.249	0.158	0.006	0.284
Receptive Language Quotient (RLQ)	0.007	−0.122	0.346 *	0.332	0.662	0.659	0.037	0.097
ADOS-Calibrated Severity Scores—Total CSS								
Social Affect—CSS	−0.111	−0.128	0.144	0.023	0.916	0.623	0.294	0.779
Repetitive Behaviour—CSS	0.125	0.002	−0.477 ***	−0.530 **	0.531	0.989	0.003	0.045
**Referred by paediatrician**	−0.085	−0.026	−0.087	−0.148	0.702	0.836	0.785	0.550

Note. Correlations are based on multiple imputed data. ^a^
*p*-values for correlation differences between groups are based on a Wald χ^2^-Test. *** *p* < 0.001, ** *p* < 0.01, * *p* < 0.05.

**Table 4 brainsci-10-00448-t004:** Regression model for age at diagnosis.

	b (SE)	β	95%-CI
**Sociodemographic Characteristics**			
Family Residence in Urban Area	1.139 (3.457)	0.023	(−0.113, 0.159)
Distance between Home and Hospital in km	−0.020 (0.038)	−0.033	(−0.156, 0.089)
Parental Level of Education above High School	−3.719 (4.103)	−0.073	(−0.231, 0.085)
Migration Status	−6.228 (3.305)	−0.136	(−0.275, 0.004)
**Clinical Characteristics**			
Male Gender	1.539 (3.433)	0.026	(−0.087, 0.138)
Non-Verbal Developmental Quotient	−0.219 * (0.098)	−0.175	(−0.336, −0.014)
Language Composite (Expressive and Receptive)	0.363 *** (0.090)	0.389	(0.204, 0.574)
ADOS-Calibrated Severity Scores—Total CSS			
Social Affect—CSS	−0.133 (0.739)	−0.013	(−0.157, 0.131)
Repetitive Behaviour—CSS	−3.575 *** (0.941)	−0.301	(−0.452, −0.150)
**Referred by paediatrician**	−9.370 ** (3.375)	−0.206	(−0.349, −0.063)
*R* ^2^	0.346		

Note. Results are based on multiple imputed data. 95%-CI = 95% confidence interval for β. *** *p* < 0.001, ** *p* < 0.01, * *p* < 0.05.

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
