# Peer review of "Explaining Age at Autism Spectrum Diagnosis in Children with Migrant and Non-Migrant Background in Austria"

_brainsci, 2020, doi:10.3390/brainsci10070448_

Round 1

Reviewer 1 Report

1. The paper writing needs to be substantially improved, introduction and conclusions are too dense and do not highlight the main points and merits of the paper.

2. Giving the study design a logistic regression would be more appropriate than comparing correlations between the two groups

3. Why do the authors claim that there is no delay in diagnosing ASD, when they observed differences in the age of diagnosis. Again, the way the paper was written is confusing.

4. Figure 2 is hard to understand. There is not a proper legend explaining the figure

5. Multiple acronymous were not explained across the text. 

Author Response

Dear Reviewer,

We would like to thank you for all your comments and suggestions that helped us improve the clarity of the manuscript. In the attachment we address all your comments individually.

Reviewer 2 Report

General issues:

This manuscript contains a comprehensive research well designed and thoroughly developed. In trying to establish whether the article contains useful information I found that this study provides significant useful contributions in order to improve assessment and early detection of ASD.

Applicability of these findings in practice is generalized to other countries, which is a valuable input of this research. Age at diagnosis of ASD is a crucial factor for efficacy of therapy; since start intervention as early as posible is a principal factor in prognosis.

COMMENTS TO THE AUTHOR

  • I would suggest include data about languages migrant parents spoke?
  • From my point of view, it looks difficult to categorize participants into 5 geographical regions, based only on primarily language used at home. It would look more properly use the country from they originally came. At least provide, in the manuscript, some more information about language/languages linked to every geographical region, in order to clarify this issue.
  • Parents from cases sample who did not have an average level of German, were excluded from the sample?
  • How could you explain/discuss the absence of Asperger in the case group? - I saw it was included in the discussion. Thanks"-
  • Children with migrant background had a lower expressive language quotient (ELQ). In discussion: could be explained because of German is not the primary language? 

I would suggest just a very few minor corrections:

  • Table 1: Please review ICD-10 ASD Score: total % sum= 100,1%
  • Line 518 : Please correct: (Asperger disorderAsperger disorder) 

Author Response

Dear Reviewer,

We appreciate your positive evaluation. We also would like to thank you for your thorough review of our manuscript. We included now data about the languages of the parents with migrant background. We also responded to your comments individually below.

Reviewer 3 Report

Dear Editor,

I have read the manuscript entitled “Explaining Age at Autism Spectrum Diagnosis in Children with Migrant and Non-Migrant Background in Austria”. 

It is a good and interesting paper overall, however, retrospective method and referral bias may be the main limitations of the study.

The authors may add a sentence in the last parts of discussion that emphasise the possible association between stress in migrant family and ASD, although they don’t have an objective measurement in the medical records.

Best Regards

Author Response

Dear Reviewer, we appreciate your positive evaluation. We have responded to your comments individually in the attachment.

Round 2

Reviewer 1 Report

The authors improved the manuscript. 

I would suggest including the 95% confidence intervals for the estimates (beta) on Table 4.

Author Response

Thank you again for your input. We appreciate your suggestion regarding table 4. Confidence intervalls are now included. We have also done another thorough spelling and style check. Kind regards
